# Characterization of a Pan-Immunoglobulin Assay Quantifying Antibodies Directed against the Receptor Binding Domain of the SARS-CoV-2 S1-Subunit of the Spike Protein: A Population-Based Study

**DOI:** 10.3390/jcm9123989

**Published:** 2020-12-09

**Authors:** Anna Schaffner, Lorenz Risch, Stefanie Aeschbacher, Corina Risch, Myriam C. Weber, Sarah L. Thiel, Katharina Jüngert, Michael Pichler, Kirsten Grossmann, Nadia Wohlwend, Thomas Lung, Dorothea Hillmann, Susanna Bigler, Thomas Bodmer, Mauro Imperiali, Harald Renz, Philipp Kohler, Pietro Vernazza, Christian R. Kahlert, Raphael Twerenbold, Matthias Paprotny, David Conen, Martin Risch

**Affiliations:** 1Landesspital Liechtenstein, Heiligkreuz, 9490 Vaduz, Liechtenstein; anna.schaffner@landesspital.li (A.S.); myriamw@bluewin.ch (M.C.W.); sarah.thiel@bluewin.ch (S.L.T.); katharina.juengert@landesspital.li (K.J.); michael.pichler@student.i-med.ac.at (M.P.); matthias.paprotny@landesspital.li (M.P.); 2Labormedizinisches Zentrum Dr Risch, Wuhrstrasse 14, 9490 Vaduz, Liechtenstein; lorenz.risch@risch.ch (L.R.); corina.risch@risch.ch (C.R.); kirsten.grossann@risch.ch (K.G.); nadia.wohlwend@risch.ch (N.W.); thomas.lung@risch.ch (T.L.); dorothea.hillmann@risch.ch (D.H.); 3Faculty of Medical Sciences, Private Universität im Fürstentum Liechtenstein, Dorfstrasse 24, 9495 Triesen, Liechtenstein; 4Center of Laboratory Medicine, University Institute of Clinical Chemistry, University of Bern, Inselspital, 3010 Bern, Switzerland; 5Cardiovascular Research Institute Basel (CRIB), University Hospital Basel, University of Basel, Spitalstrasse 2, 4056 Basel, Switzerland; stefanie.aeschbacher@usb.ch (S.A.); raphael.twerenbold@usb.ch (R.T.); 6Labormedizinisches Zentrum Dr Risch, Waldeggstrasse 37, 3097 Liebefeld, Switzerland; susanna.bigler@risch.ch (S.B.); thomas.bodmer@risch.ch (T.B.); 7Centro Medicina di Laboratorio Dr Risch, Via Arbostra 2, 6963 Pregassona, Switzerland; mauro.imperiali@risch.ch; 8Institute of Laboratory Medicine and Pathobiochemistry, Molecular Diagnostics, University Hospital Giessen and Marburg, Philipps University Marburg, Baldingerstraße, 35043 Marburg, Germany; Harald.Renz@uk-gm.de; 9Department of Infectious Diseases and Hospital Epidemiology, Cantonal Hospital St. Gallen, Rohrschacherstrasse 95, 9007 St. Gallen, Switzerland; philipp.kohler@kssg.ch (P.K.); pietro.vernazza@kssg.ch (P.V.); christian.kahlert@kssg.ch (C.R.K.); 10Department of Infectious Diseases and Hospital Epidemiology, Children’s Hospital of Eastern Switzerland, Claudiusstrasse 6, 9006 St. Gallen, Switzerland; 11Clinic of Cardiology, University Hospital Basel, Petersgraben 4, 4031 Basel, Switzerland; 12Population Health Research Institute, McMaster University, 237 Barton Street East, Hamilton, ON L8L 2X2, Canada; conend@mcmaster.ca; 13Central Laboratory, Kantonsspital Graubünden, Loësstrasse 170, 7000 Chur, Switzerland

**Keywords:** area under the curve, coronavirus, COVID-19, diagnostic accuracy, immunoassay, predictive values, SARS-CoV-2, sensitivity, serology, specificity

## Abstract

Pan-immunoglobulin assays can simultaneously detect IgG, IgM and IgA directed against the receptor binding domain (RBD) of the S1 subunit of the spike protein (S) of severe acute respiratory syndrome coronavirus 2 (SARS-CoV-2 S1-RBD Ig). In this work, we aim to evaluate a quantitative SARS-CoV-2 S1-RBD Ig electrochemiluminescence immunoassay (ECLIA) regarding analytical, diagnostic, operational and clinical characteristics. Our work takes the form of a population-based study in the principality of Liechtenstein, including 125 cases with clinically well-described and laboratory confirmed SARS-CoV-2 infection and 1159 individuals without evidence of coronavirus disease 2019 (COVID-19). SARS-CoV-2 cases were tested for antibodies in sera taken with a median of 48 days (interquartile range, IQR, 43–52) and 139 days (IQR, 129–144) after symptom onset. Sera were also tested with other assays targeting antibodies against non-RBD-S1 and -S1/S2 epitopes. Sensitivity was 97.6% (95% confidence interval, CI, 93.2–99.1), whereas specificity was 99.8% (95% CI, 99.4–99.9). Antibody levels linearly decreased from hospitalized patients to symptomatic outpatients and SARS-CoV-2 infection without symptoms (*p* < 0.001). Among cases with SARS-CoV-2 infection, smokers had lower antibody levels than non-smokers (*p* = 0.04), and patients with fever had higher antibody levels than patients without fever (*p* = 0.001). Pan-SARS-CoV-2 S1-RBD Ig in SARS-CoV-2 infection cases significantly increased from first to second follow-up (*p* < 0.001). A substantial proportion of individuals without evidence of past SARS-CoV-2 infection displayed non-S1-RBD antibody reactivities (248/1159, i.e., 21.4%, 95% CI, 19.1–23.4). In conclusion, a quantitative SARS-CoV-2 S1-RBD Ig assay offers favorable and sustained assay characteristics allowing the determination of quantitative associations between clinical characteristics (e.g., disease severity, smoking or fever) and antibody levels. The assay could also help to identify individuals with antibodies of non-S1-RBD specificity with potential clinical cross-reactivity to SARS-CoV-2.

## 1. Introduction

The diagnosis of acute coronavirus disease 2019 (COVID-19) with laboratory parameters relies on RT-PCR or antigen tests demonstrating the presence of severe acute respiratory syndrome coronavirus 2 (SARS-CoV-2) virus in the sampled material; e.g., in nasopharyngeal swabs [1]. In patients with a high clinical suspicion of COVID-19 with a negative RT-PCR result for SARS-CoV-2, demonstrating the presence of antibodies against SARS-CoV-2 can resolve a diagnostic dilemma and overcome the problem of false negative RT-PCR [2]. Because SARS-CoV-2 infection has also been shown to have an asymptomatic or oligosymptomatic course, antibodies against SARS-CoV-2 have proven to be a valuable tool to assess the seroprevalence of SARS-CoV-2 infection either in the total population, in healthcare workplace settings or in general workplace settings [3,4,5]. Serological testing facilitates surveillance and assists in the identification of individuals susceptible to COVID-19 infection [6,7]. The serological analysis of convalescent SARS-CoV-2 antibodies also allows for the identification of individuals that are potentially suited to serve as plasma donors of convalescence plasma, which has been proposed to aid in the therapy of severe cases of COVID-19 [8,9,10]. Further, many individuals have not had access to the molecular diagnosis of COVID-19 with RT-PCR [11]. The serological analysis of SARS-CoV-2 antibodies can help to clarify whether these individuals have had SARS-CoV-2 infection in the past [12]. At present, there is uncertainty regarding whether the presence of SARS-CoV-2 antibodies confers protection against reinfection, and there is conflicting evidence regarding whether antibodies against SARS-CoV-2 can persist over a longer period of time [13,14].

A multitude of assays is available to measure different isotypes of different antibody specificities against SARS-CoV-2 with different analytical techniques [15]. Regarding isotypes, assays usually measure total antibodies (i.e., combined IgA, IgM, and IgG), IgA, IgM and IgG [12]. Analytical methods comprise chemiluminescence assays (CLIA), enzyme linked immunosorbent assays (ELISA) or immunochromatographic lateral flow assays. Regarding epitope specificities, antibodies are directed against the nucleocapsid (N) antigen or against the spike (S) protein [16]. The spike protein has two subunits, S2 and S1, and the S1 subunit carries the receptor binding domain (RBD) [17]. The RBD of the virus binds to the human surface-expressed angiotensin-converting enzyme 2 (ACE2) which leads to the internalization and infection of human cells [18]. Antibodies against S1-RBD have been shown to confer a neutralizing activity against SARS-CoV-2 [19]. Although not possessing neutralizing activity themselves, antibodies against S1/S2 dimer or non-S1-RBD specificity have been shown to correlate with neutralizing antibody titers to a weak to moderate extent [20,21].

Recently, a novel assay for the quantitative measurement of total antibodies against SARS-CoV-2 S1-RBD with electrochemiluminescence (ECLIA, Roche Diagnostics, Rotkreuz, Switzerland) has become available. In this work, we aim to determine the diagnostic (i.e., sensitivity and specificity), operational (i.e., positive and negative predictive values) and clinical (i.e., associations with clinical signs and symptoms) characteristics of this assay in a population-based setting in the Principality of Liechtenstein. As a secondary objective, we describe the analytical characteristics of this novel assay (imprecision, linearity, analytical specificity).

## 2. Methods

### 2.1. Study Participants

In this work, we consider a diagnostic test measuring total SARS-CoV-2 antibodies directed at the RBD-domain of the S1 subunit of the spike protein (SARS-CoV-2 S1-RBD Ig) in serum. In order to evaluate the diagnostic sensitivity and diagnostic specificity, two cohorts were assembled in the principality of Liechtenstein, a small European country situated between Switzerland and Austria with a population of 38,749 inhabitants. The first cohort (COVID-FL) consisted of individuals who were investigated for SARS-CoV-2 infection during the first wave of the pandemic, which lasted from 2 March until 23 April 2020, and who had follow-up samples for serological SARS-CoV-2 antibody testing available. A detailed description of the cohort is provided in [4,22]. In brief, this first cohort consisted of all of the country’s COVID-19 cases, their household contacts, and their close working contacts (*n* = 261). Most of the patients and close contacts (248/261; i.e., 95%; 95% confidence interval, CI, 91.7–97) had at least one follow-up sample for serological investigations available. Patients with SARS-CoV-2 infection had their first follow-up sample taken after a median of 48 days (interquartile range, IQR, 43–52) after symptom onset, and 114 of these 125 cases (i.e., 91.2%, 95% CI, 84.9–95) also provided a second follow-up sample with a median of 139 days (interquartile range, IQR, 129–144) after symptom onset.

The second cohort consisted of participants from the population-based prospective COVI-GAPP study, which aims to define the role of a sensory bracelet for improved early recognition of COVID-19 within the COVID-RED (COVID-19 remote early detection) consortium [23]. For the present study, the baseline serological sample was taken between 2 June and 6 October. During this timeframe, the country registered 48 cases corresponding to a low incidence of 123 cases per 100,000 inhabitants within 4 months (mean weekly incidence 7 per 100,000 inhabitants).

In order to study the analytical specificity, we further included anonymized historical samples originating from Swiss- and Liechtenstein-based patients that were known to have an active or reactivated specific viral disease (Epstein-Barr virus (EBV), *n* = 8; cytomegalovirus (CMV), *n* = 7; other endemic common-cold coronaviruses (HKU1, NL63, OC43, 229E), *n* = 12) to explore any cross-reactivity causing false positive results in the SARS-CoV-2 serology. Endemic coronavirus disease was diagnosed with a FilmArray multiplex PCR System (BioFire, BioMérieux, Petit Lancy, Switzerland) during 2019 in ten cases, in January 2020 in one case, and in mid-February 2020 in one case, eight days before the first case of COVID-19 was reported in Switzerland. The last serum sample of a patient with endemic coronavirus disease was collected on 2 March 2020, which was seven days after the first case in Switzerland was identified. Samples from patients with active EBV (VCA IgM positive, EBNA IgG negative) as well as active or reactivated CMV infection (IgG positive, IgM positive) were all drawn in 2019; i.e., the pre-pandemic era in Switzerland and Liechtenstein. The sera from patients with preceding common cold coronavirus disease and CMV and EBV infection were stored at −25 °C. The characteristics of the cohorts providing samples included in this study are summarized in Table 1.

The study protocol was verified by the cantonal ethics boards of Zurich (BASEC Req-20-00786 and Req-20-00676) and Eastern Switzerland (EKOS; BASEC Nr. Req-20-00586). Study participants in both Liechtenstein cohorts provided written informed consent, while informed consent for performing laboratory analysis on anonymized samples in the third cohort was waived.

### 2.2. Data Collection

Patients from the COVID-FL cohort as well as their household and close working contacts were closely followed every two to three days by telephone until recovery, and demographic, anthropometric and clinical data were collected. The details of this cohort have been described in detail in [4]. The individuals of this cohort had venous blood drawn for serological analysis for SARS-CoV-2 antibodies with different test formats. These test formats comprised chemiluminescence immunoassays (CLIA; i.e., chemiluminescent microparticle immunoassay, CMIA, (IgG with anti-N-specificity) (Abbott Diagnostics Baar, Switzerland); electrochemiluminescence, ECLIA, (pan immunoglobulin with anti-N-specificity) (Roche Diagnostics, Rotkreuz, Switzerland); and luminescence immunoassay, LIA, (IgG with anti-S1/S2-specificity; IgM with anti-S1/S2-specificity, only in the subset of COVID-19 index cases due to restricted reagents) (Diasorin, Luzern, Switzerland)); enzyme linked immunosorbent assays, ELISA, measured with reagents from Euroimmun (Luzern, Switzerland; IgG and IgA isotypes with anti-S1-protein specificity), Epitope Diagnostics (Bencard, Greifensee, Switzerland; IgG and IgM isotypes with anti-N-antigen specificity) and a lateral flow assay (Sugentech, Daejeon, Republic of Korea; SGTi-flex COVID-19 IgM/IgG measuring IgG and IgM isotypes with anti-N-antigen specificity). Some of the results of the evaluation of these other assays have been partly published or deposited elsewhere [22,24]. As the test under investigation was not available at that time, we used materials stored at −80 °C for this study. Measurements were performed in October 2020.

Individuals from the second cohort without SARS-CoV-2 infection were asked whether they had tested positive for SARS-CoV-2 infection with RT-PCR, and a venous serum sample was included at baseline. These samples were tested with the same SARS-CoV-2 antibody assays as the individuals of the COVID-FL cohort, except for the Epitope Diagnostics ELISAs, which were discontinued in our laboratory due to stability problems after about 80% of the cohort had been tested. The majority of measurements were done in October 2020 on sample materials stored at −25 °C. A few samples were tested on fresh material, as the assay was operational when the samples were drawn.

The presence of SARS-CoV-2 infection was adjudicated based on RT-PCR tests performed on a Roche Cobas 6800 (Roche Diagnostics, Rotkreuz, Switzerland) [2,25] and serology [24]. A patient was considered to have had SARS-CoV-2 infection if either a positive RT-PCR result was found and/or two serological assays were positive: one detecting antibodies against N-antigen (ECLIA) and one detecting antibodies against S1/S2-antigen (LIA) and/or S1-antigen IgG (ELISA). Cut-off indices for the positivity of the employed serological assays were taken from the manufacturer: i.e., ≥1 for ECLIA, ≥15 for LIA, and ≥1.1 for ELISA [2].

The sera employed to assess analytical specificity were recovered from stored sera sent to the Labormedizinisches zentrum Dr Risch Ostschweiz AG in Switzerland, a medical laboratory accredited according to ISO 17025. It is common practice in this laboratory to store biological material for at least 13 months in order to allow for lookback analyses for several indications. These sera were stored at −25 °C.

## 3. Laboratory Analysis

Serum was measured on a COBAS 6000 instrument (Roche Diagnostics, Rotkreuz, Switzerland) with the quantitative Elecsys^®^ Anti-SARS-CoV-2 S antibody assay (pan-SARS-CoV-2 S1-RBD Ig). This assay is calibrated with a two-point calibration and quantifies antibodies directed against the receptor binding domain (RBD) of the SARS-CoV-2 spike protein. The assay is a one-step double antigen sandwich assay employing biotinylated RBD-antigen and ruthenylated RBD-antigen. In the presence of SARS-CoV-2 S1-RBD antibodies, double antigen sandwich (DAGS) immune complexes are formed. After the addition of streptavidin-coated microparticles, the DAGS can be immobilized, washed and brought to detection with ECLIA in the measuring cell. Increasing antibody concentrations induces increasing measurement signals. On the COBAS 6000, a 20 µL sample is used for analysis. The manufacturer reports a linear measurement range from 0.4 to 250 U/mL with a manufacturer cut-off of 0.8 U/mL and higher, indicating a reactive or positive sample. For statistical calculations, results lower than 0.4 U/mL were assumed to be 0.39 U/mL, and results higher than 250 U/mL were assumed to be 250.1 U/mL.

### 3.1. Assay Validation

Imprecision was assessed by inter and intra-assays at two different concentrations. For this purpose, we employed pooled materials and commercial control materials supplied by the manufacturer of the assay. Precision studies were done by repeating measurements 10 times during one day and daily for 10 days. Further, we assessed linearity in a serum pool displaying an antibody concentration of 105 U/L by serially diluting the material with the kits’ specific diluent at dilutions of 1:2, 1:4, 1:8, 1:16, 1:32, 1:64 and 1:128. We did not assess interference by lipemia and bilirubinemia. We finally compared the findings of the pan-SARS-CoV-2 S1-RBD Ig assay to the results of the SARS-CoV-2 S1 ELISA (IgG and IgA) and the SARS-CoV-2 S1/S2 LIA (IgG and IgM).

### 3.2. Statistical Methods

Continuous variables were given as medians and interquartile ranges (IQRs), whereas proportions were given as percentages together with 95% confidence intervals (CIs). Agreement between two methods was assessed by Spearman rank correlation and Passing Bablok regression [22]. In order to determine the concordance of different antibody assays against SARS-CoV-2 spike protein (pan-SARS-CoV-2 S1-RBD Ig, SARS-CoV-2 S1/S2 IgG, and SARS-CoV-2 S1 IgG/IgA), Venn diagrams were drawn [26]. Continuous variables between the two groups were compared with the Mann–Whitney U test, whereas proportions were compared by means of the Chi-square test. For comparison of medians of the three groups, the Kruskal–Wallis test was employed, and the Jonckshere–Terpstra test was used to investigate linear trends. Diagnostic specificity was determined in the samples originating from the COVID-19-negative participants of the COVI-GAPP study as well as in household contacts and close working contacts without evidence of SARS-CoV-2 infection. Diagnostic sensitivity was assessed in the COVID-19 cases of the COVID-FL cohort. Receiver operating characteristic (ROC) curves with the area under the curves (AUCs) were calculated as an indicator of diagnostic accuracy. The AUCs for anti-S-assays were compared by the method of Hanley and McNeil. Positive and negative predictive values for each of the employed assays were then plotted as a function of pretest probability, as described elsewhere [27]. The positive predictive value from a combined orthogonal testing algorithm with pan-SARS-CoV-2 N-antigen Ig and pan-SARS-CoV-2 S1-RBD Ig was also calculated [8,28]. The calculator provided by the U.S. American Food and Drug administration was employed for this purpose [28]. Finally, we compared the antibody titers of the pan-SARS-CoV-2 S1-RBD Ig in the first and second serological measurements in order to detect a potential kinetic over time. *p*-values < 0.05 were considered statistically significant. MedCalc version 18.11.3 (Mariakerke, Belgium), GraphPad 8.4.3. (686) (GraphPad Software LLC., San Diego, CA, USA) and Microsoft Excel 2016 MSO (16.0.8431.2046) (Microsoft Inc., Seattle, WA, USA) were used for statistical and graphical computations.

## 4. Results

### 4.1. Baseline Characteristics

The COVID-FL cohort consisted of 265 individuals, 248 (121 female/127 male; mean age 41 years, IQR, 28–55) of which serum was available. In total, 95 of the COVID-FL cohort were COVID-19 index cases with positive RT-PCR results. Of these, 90 had serum samples available for antibody testing in the first follow-up median 48 days (IQR, 43–52) after symptom onset, and 82 had samples available at the second follow-up median 140 days (IQR, 133–145) after symptom onset. During acute disease, 11 of the index patients were hospitalized; none of them needed intensive care, but one of the hospitalized patients died at age 94. Of the 170 households (*n* = 109) and close working contacts (*n* = 61), 158 had samples available for antibody testing in the first follow-up. Of these, 35 presented evidence for SARS-CoV-2 infection by combined serological results. In these 35 cases, serum was taken 46 days (IQR, 43–53) after symptom onset (estimated symptom onset in asymptomatic cases based on symptom onset of index cases plus estimated seven days of incubation period) and a follow-up sample was available in 31 cases of them (with a median time after symptom onset of 139 days, IQR, 129–146). Of the close contacts with COVID-19 cases, one was hospitalized, and nine were asymptomatic. In the cohort recruited for the COVI-GAPP study, 1063 individuals participated. Of these, twelve had serological evidence of SARS-CoV-2 infection and, with one asymptomatic exception (a household contact of a COVID-19 case), were symptomatic during the first wave. These twelve cases and 3 participants without sufficient sample material for testing of pan-SARS-CoV-2 S1-RBD Ig were therefore excluded from further analysis, leaving 1048 individuals (621 females/427 males; median age 45 years, IQR, 39–48) available for the evaluation of specificity. This represents 9.7% of the entire population of the respective age stratum (*n* = 10,830) in the Principality of Liechtenstein [29]. The results for pan-SARS-CoV-2 S1-RBD Ig levels are shown for patients with and patients without SARS-CoV-2 infection (Figure 1). Samples employed for the determination of analytic specificity originated from 27 individuals (17 females/10 males; median age 39 years, IQR, 17–60).

### 4.2. Assay Validation

The inter-series coefficient of variation (CV) in the serum was 3.0% at an antibody concentration of 10.2 U/mL (*n* = 10; positive test kit control material) and 1.3% at an antibody concentration of 24.5 U/mL (*n* = 10; pooled serum). The intra-series CV’s were 1.3% (*n* = 10; positive control) and 0.7% (*n* = 10; pooled serum). Serial dilutions revealed a linear curve, as shown in Figure 2. There was a close correlation between SARS-CoV-2 S1/S2 IgG and pan-SARS-CoV-2 S1-RBD Ig (*r* = 0.97, 95% CI, 0.96–0.97; Passing-Bablok regression line SARS-CoV-2 S1-RBD Ig = −5.5 + 1.6 × SARS-CoV-2 S1/S2 IgG) as well as between SARS-CoV-2 S1 IgG and pan-SARS-CoV-2 S1-RBD Ig (*r* = 0.90, 95% CI, 0.88–0.93; SARS-CoV-2 S1-RBD Ig = −4.1 + 22.6 × SARS-CoV-2 S1 IgG). The correlation was somewhat weaker between SARS-CoV-2 S1 IgA and pan-SARS-CoV-2 S1-RBD Ig (*r* = 0.81, 95% CI, 0.76–0.85; SARS-CoV-2 S1-RBD Ig = −7.9 + 27.7 × SARS-CoV-2 S1 ELISA IgA) as well as SARS-CoV-2 S1/S2 IgM and pan-SARS-CoV-2 S1-RBD Ig (*r* = 0.5, 95% CI, 0.35–0.62; SARS-CoV-2 S1-RBD Ig = −136 + 87.8 × SARS-CoV-2 S1/S2 IgG).

### 4.3. Analytic Specificity

Serum samples of patients with endemic common cold coronavirus had their samples taken a median of 94 days (IQR, 30–235) after diagnosis. Of the sera taken after endemic common-cold coronavirus infection, four were infected with RC229E, three were infected with RCNL63, two were infected with RCHKU1, two had an infection with RCOC43, and one patient had both RC229E and RCNL63. None of these patients with common-cold coronavirus infection had measurable antibody concentrations for pan-SARS-CoV-2 S1-RBD Ig (results were <0.4 U/mL). Six of the eight patients with acute EBV disease showed the presence of heterophilic antibodies. None of the patients with active EBV or CMV disease exhibited measurable antibody concentrations for pan-SARS-CoV-2 S1-RBD Ig (all results <0.4 U/mL).

### 4.4. Diagnostic Specificity and Sensitivity at the Manufacturers’ Cutoff

Of the 125 cases with SARS-CoV-2 infection from the COVID-FL cohort with serum available for the determination of pan-SARS-CoV-2 S1-RBD Ig on the occasion of the first serological follow-up, 122 had a positive serum result above the manufacturers’ cut-off. At the manufacturers’ cut-off, this translates into a diagnostic sensitivity of 97.6% (95% CI, 93.2–99.1) at the first follow-up investigation. Interestingly, the three negative results also had negative results for all the other assays. At the second serological follow-up, 112 of 114 cases with SARS-CoV-2 infection had a positive serological result translating into a comparable sensitivity of 98.2% (95% CI, 93.9–99.5). Two of the patients with negative pan-SARS-CoV-2 S1-RBD Ig belonged to the negative patients on the first follow-up and also had negative results on all of the other tests.

In the subgroup of household contacts and close working contacts without evidence of COVID-19 in the COVID-FL cohort, one of 123 participants had positive pan-SARS-CoV-2 S1-RBD Ig results (3.1 U/mL), translating into a specificity of 99.2% (95% CI, 95.6–99.8). This participant was shown to be negative for all other serological tests. Of the 1048 participants of the COVI-GAPP study without evidence of SARS-CoV-2 infection, 1036 had blood drawn and sample material available for the testing of pan-SARS-CoV-2 S1-RBD Ig. One of the participants had a positive SARS-CoV-2 S1-RBD Ig (1.0 U/mL) with all other assays displaying clearly negative results. This translates into a specificity of 99.9% (95% CI, 99.5–100). A consolidated analysis of both cohorts (*n* = 1159) revealed two participants with positive pan-SARS-CoV-2 S1-RBD Ig results, translating into a specificity of 99.8% (95% CI, 99.4–99.9).

In order to determine the diagnostic accuracy of the pan-SARS-CoV-2 S1-RBD Ig assay regarding the diagnosis of SARS-CoV-2 infection, a receiver operating characteristic (ROC) analysis was done on the combined dataset of the COVI-GAPP study and the COVID-FL cohort (i.e., 125 cases with SARS-CoV-2 infection, 1159 individuals without evidence of SARS-CoV-2 infection). The area under the curve (AUC) at the first follow-up was 0.984 (95% CI, 0.976–0.99) (Figure 3). The AUC did not substantially change when analyzing the results of the second follow-up (0.988, 95% CI, 0.979–0.993).

There were no significant differences between the area under the curves of pan-SARS-CoV-2 S1-RBD Ig (AUC 0.984, 95% CI, 0.976–0.99), SARS-CoV-2 S1/S2 IgG (AUC 0.98, 95% CI, 0.971–0.987), and SARS-CoV-2 S1 IgG (AUC 0.979, 95% CI, 0.970–0.986). However, the AUC of SARS-CoV-2 S1 IgA (AUC 0.941, 95% CI, 0.926–0.953) was significantly lower than that of the other three assays (*p* < 0.01 for all).

### 4.5. Association of Pan-SARS-CoV-2 S1-RBD with Clinical Variables

We then investigated whether clinical symptoms were associated with pan-SARS-CoV-2 S1-RBD antibody titers. We found significant differences in antibody titers between symptomatic hospitalized patients, symptomatic outpatients and asymptomatic patients (*p* < 0.001) (Figure 4). A significant linear trend with declining antibody titers in less severely sick SARS-CoV-2 infection was observed (*p* < 0.001). The same differences could be found at the second follow-up.

After a median of 48 days following symptom onset in cases with SARS-CoV-2 infection, fever was the only symptom associated with significantly different antibodies (Table 2). Interestingly, patients who smoked tended to have lower antibody titers on the first follow-up than nonsmoking patients (*p* = 0.05). At the second follow up, smokers had significantly lower antibody levels than non-smokers (31, IQR, 18–121, vs. 112 U/mL IQR, 51–242; *p* = 0.04) despite the fact that disease duration did not differ between the two groups (15 days, IQR, 3–19) in smokers vs. in nonsmokers; (10 days, IQR, 5–16; *p* = 0.26).

### 4.6. Kinetics of SARS-CoV-2 S1-RBD Antibodies

Of the 125 cases with SARS-CoV-2 infection in the COVID-FL cohort, 114 had a second follow-up sample taken after a median of 139 days. Pan-SARS-CoV-2 S1-RBD Ig in levels at the first follow-up was a median of 66 U/L (IQR, 25–174). Interestingly, the antibody levels significantly increased to the second follow-up to a median of 109 U/L (IQR, 46–227) (*p* < 0.001). The course of the antibody levels is shown in Figure 5. Of the patients without pan-SARS-CoV-2 S1-RBD Ig higher than 250 U/L at both blood drawings (*n* = 99), 68 displayed an increase, 27 had a decrease and 4 showed unchanged antibody levels.

### 4.7. Sensitivity of Other Anti-Spike Protein Antibodies in Patients with SARS_CoV-2 Infection

Of the 125 individuals with SARS-CoV-2 infection in the COVID-FL cohort, three did not display a pan-SARS-CoV-2 S1-RBD antibody response, when applying the manufacturer’s cutoffs. These patients neither responded to other anti-S specificities (S1/S2; non-S1-RBD) nor had antibodies against N-antigen. Further, 14 patients (11%, 95% CI, 7–18) with SARS-CoV-2 infection displayed negative results in one or more assays directed against non-S1-RBD epitopes of the spike protein, although they displayed positive pan-SARS-CoV-2 S1-RBD antibodies (Figure 6). The titer of these pan-SARS-CoV-2 S1-RBD antibodies in these 14 patients, however, was significantly lower than that in the patients with non-S1-RBD antibodies positive in all three other assays (6.9, IQR, 3–12.9, vs. 79.9, IQR, 44.2–183) U/mL; *p* < 0.001). Together, when using manufacturer’s cutoffs for non-S1-RBD assays, about 11% of SARS-CoV-2 infection patients lacked antibodies with RBD-S1 specificity at a moderate antibody level.

### 4.8. Specificity of Anti-Spike Protein Antibodies in Patients without SARS-CoV-2 Infection

We further detailed the specificity of antibody test formats determining antibodies against spike protein: the LIA detects antibodies directed against S1 and S2-subunits of the spike protein, the ELISA IgG and IgA detect antibodies against non-RBD epitopes of the S1 subunit of the spike protein, while the ECLIA detects antibodies against the RBD of the S1 epitope. For this analysis, we included individuals from the COVI-GAPP and the COVID-FL cohort. A complete set of serology results was available for 1159 individuals. When applying the manufacturers’ cut-offs, 90 individuals (7.8%, 95% CI, 6.4–9.5) had positive antibody responses with one or more assays (Figure 7). A vast majority of these 90 individuals showed non-S1-RBD antibody positivity (i.e., 64/90, 71%, 95% CI, 61–79) in single assays. From the other 26 individuals, however, only two had a positive result in the SARS-CoV-2 S1/S2 IgG assay as well as in a non-S1-RBD assay (IgG or IgA), indicating reactivity to non-S1-RBD epitopes on the S1-subunit of the spike protein. Another 16 samples had SARS-CoV-2 S1/S2 IgG positive results without positivity in the non-S1-RBD assays, indicating antibodies against S2. Furthermore, there were two samples with isolated pan-SARS-CoV-2 S1-RBD antibodies, indicating another isotype of S1-specificity or lower titers of non-S1-RBD antibodies below the manufacturers’ cutoffs.

We then applied half of the manufacturers’ cutoff as a decision criterion for antibody positivity (Figure 8) [24]. Interestingly, with these modified cutoffs, the two pan-SARS-CoV-2 S1-RBD Ig positive signals also shared measurement signals with other tests, i.e., from the non-S1-RBD-isotype. With the modified cut-offs, there were 250 individuals (21.6%, 95% CI, 19.3–24.1) with a result above the cut-off. The overlap between non-RBD anti-S1/S2 and anti-S1 antibodies (*n* = 18) increased. Together, individuals without evidence of SARS-CoV-2 infection in about 8% to 22% (depending on the cut-offs) of cases had antibody results directed against different domains of the SARS-CoV-2 spike protein.

### 4.9. Positive and Negative Predictive Values

When taking the specificity (99,8%) and sensitivity (97.6%) of the pan-SARS-CoV-2 S1-RBD Ig assay, positive (PPV) and negative predictive values (NPV) can be calculated as a function of the pretest probability of past SARS-CoV-2 infection (Table 3).

When only employing a pan-SARS-CoV-2 S1-RBD result, PPVs with a pretest probability of prior SARS-CoV-2 infection became 95% and higher at pretest probabilities of 5% and higher. Confirming a positive result in an orthogonal testing algorithm with a positive anti-N-antigen ECLIA result, which had a sensitivity of 96% and a specificity of 99.9% [24], revealed a PPV of 100% even at a pretest probability of 1%. A negative pan-SARS-CoV-2 S1-RBD Ig result up to a pretest probability of 40% resulted in an NPV of 98% or higher. Using an orthogonal testing algorithm for negative results leads to worse NPV scores than those obtained for testing with one assay only. The PPVs and NPVs for all possible pretest probabilities at the manufacturers’ cutoffs as well as half and double of the manufacturers’ cutoffs are provided in Figure 9a (PPV) and Figure 9b (NPV). In our cohorts, the pan-SARS-CoV-2 S1-RBD Ig had a sensitivity (i.e., 97.6%, 95% CI, 93.2–99.1) and specificity (i.e., 99.8%, 95% CI, 99.4–99.9) at half of the manufacturer’s cutoff index (i.e., COI > 0.4) identical to the diagnostic characteristics observed at the manufacturer’s cutoff. At double the manufacturer’s cut-off (i.e., COI > 1.6) the sensitivity was 96.8% (95% CI, 92.1–98.7), whereas the specificity amounted to 99.9% (95% CI, 99.5–99.98).

## 5. Discussion

The present population-based study investigated the analytical (analytical specificity), diagnostic (diagnostic sensitivity and specificity) and operational characteristics (predictive values) of a quantitative assay for the determination of total antibodies directed against the receptor binding domain (RBD) of the spike protein S1 subunit—the so-called pan-SARS-CoV-2 S1-RBD antibody assay. The assay showed good analytic and diagnostic characteristics. The antibody titers exhibited associations with clinical characteristics, such as fever (positively associated) or smoking (inverse association). Pan-SARS-CoV-2 S1-RBD antibody levels in cases with SARS-CoV-2 infection were demonstrated to remain sustainable and increased for at least five months. Finally, the investigation of the specificity of antibodies directed against different parts of the spike protein revealed considerable proportions of patients in a population-based sample who exhibited anti-spike protein reactivity without RBD-specificity.

The investigated assay is a pan-immunoglobulin assay measuring IgG, IgM and IgA isotypes. It correlates well with other anti-spike protein antibody assays with IgG isotypes but less well with IgA and IgM isotypes. The assay displays a linear measurement signal behavior over the whole measurement range from 0.4 to 250 U/L. Quantitative measurements are possible up to 313 times the manufacturer’s cutoff, allowing the assay to capture titer dynamics within a broad band in the positive range. Analytically, no interference conferring a diminished specificity could be observed with heterophilic antibodies or individuals with other common cold coronavirus diseases in their history. In summary, when also taking into account the imprecision specifications, the assay can be regarded to offer solid analytical characteristics for determining, quantifying and monitoring SARS-CoV-2 S1-RBD Ig.

To the best of our knowledge, this is the first paper to describe this novel assay in a large validation study. We found that the diagnostic accuracy of the pan-SARS-CoV-2 S1-RBD Ig assay is comparable to other anti-spike protein assays determining specific IgG, with the latter assays also having been found to have similar diagnostic accuracies in other studies [30,31]. However, the evaluated assay was substantially better than the diagnostic accuracy found for SARS-CoV-2 S1 IgA, which is in agreement with the consensus that IgA should not be used in routine serological analyses to detect evidence of past SARS-CoV-2 infection [32]. Whereas the pan-SARS-CoV-2 S1-RBD Ig showed favorable operational characteristics in excluding a past SARS-CoV-2 infection (i.e., an NPV ≥ 99% up to pretest probabilities of 30%), good positive predictive values were achieved with an orthogonal testing approach employing two pan-SARS-CoV-2 assays—i.e., one with N-antigen specificity and one with S1-RBD specificity—in agreement with the findings of Gudbjartsson et al. [33].

It has commonly been stated that specific antibodies decline after SARS-CoV-2 infection and that potentially protective immunoglobulins will wane over a relatively short time [13,34,35,36,37]. These studies have been done by using isotype-specific assays. Somewhat in contrast to these investigations, we were able to demonstrate that pan-SARS-CoV-2 N-antigen antibody remained constant over a duration of five months after the onset of SARS-CoV-2 infection [14]. In line with our findings, another population-based study in Iceland demonstrated that pan-SARS-CoV-2 N-antigen and pan-SARS-CoV-2 S1-RBD antibody positivity rates remained constant over 112 days [33]. In our cohort, the first follow-up samples were taken after about seven weeks, whereas the second follow-up samples were taken nearly five months after the onset of symptoms. We could observe an increase of pan-SARS-CoV-2 S1-RBD antibody titers after the first follow-up. This is in accordance with the findings of Gudbjartsson and colleagues, who showed with pan-SARS-CoV-2 assays that titers increased for two months and then reached a plateau for at least another two months [33]. Our findings are confirmative in this regard and demonstrate that the plateau of pan-SARS-CoV-2 S1-RBD antibody levels in contrast to the declining isotype specific SARS-CoV-2 antibodies is maintained at least for another month [33]. Our findings also are in line with the observation that pan-SARS-CoV-2 N-antigen antibody levels were increased after 6.2 months [38]. In summary, it can be postulated that pan-SARS-CoV-2 S1-RBD and pan-SARS-CoV-2 N-antigen antibody levels show sustained levels up to half a year, even if isotypes exhibit a decline during the same time. Regarding vaccination, we consider these findings as promising because they suggest a sustained humoral immune response to viral proteins.

The fact that the severity of COVID-19 is correlated with antibody levels after the recovery of the disease is known [38]. We can show that, of all the recorded COVID-19 symptoms, fever during COVID-19 is the main determinant of antibody levels in due course. We were somewhat surprised by the fact that smokers had lower antibody levels than nonsmokers. However, such a phenomenon has already been observed in other viral diseases, such as Human Papilloma Virus (HPV) infection [39,40]. Even if smokers seem to have a lower risk of being infected with SARS-CoV-2 [41], it has been shown that smoking is a predictor of mortality [42]. In contrast to anti-N and anti-S1/S2 antibodies, antibodies against S1-RBD have been demonstrated to closely correlate with neutralizing activity of sera [19,20,21,43]. Despite this, since we did not assess neutralizing activity of pan-SARS-CoV-2 S1-RBD positive sera, there remains some uncertainty how closely the pan-SARS-CoV-2 S1-RBD correlates with neutralizing antibodies. Assuming that the pan-SARS-CoV-2 S1-RBD correlates with neutralizing antibodies and assuming that higher antibody levels may confer better protection from infection, it could be hypothesized that (a.) the lower risk of COVID-19 in smokers is probably not due to a weaker antibody response and (b.) that higher COVID-19 mortality in smokers could inter alia occur due to an impaired humoral immune response. Assuming further that lower antibody levels may confer a higher risk of reinfection, it could be hypothesized that smokers could be at a higher risk of reinfection for COVID-19. However, our observation was obtained in a small sample of only eleven smokers, which did not allow our findings to be controlled for other factors. Accordingly, our findings warrant confirmation in other, larger study settings.

Among the different commercially available assays measuring antibodies against spike protein, the pan-SARS-CoV-2 S1-RBD assay showed the fewest false positive results. This assay thus seems to be the most specific regarding the diagnosis of a past SARS-CoV-2 infection. Assays with specificity to antibodies targeting other epitopes of the spike protein (i.e., non-S1-RBD, non-RBD S1/S2) may be able to capture cross-reacting antibodies, which to some extent could confer immune competence against SARS-CoV-2 infection. Our study demonstrates that such non-S1-RBD antibody reactivity against spike protein frequently occurs (i.e., 248/1159, 21.4% 95% CI (19.1–23.9)) in patients without SARS-CoV-2 infection, especially if cutoffs are lowered to half of the manufacturer’s cut-offs. Presumably, the manufacturers set the cut-offs to avoid a decrease of assay specificity due to cross-reacting antibodies. Ng and colleagues identified such cross-reacting antibodies against non-RBD S-protein epitopes in sera originating from the pre-pandemic era, which also demonstrated activity in the neutralization assay [44]. Cross-reacting antibodies due to endemic common cold coronavirus disease have also been demonstrated by others [45,46,47,48,49]. It remains to be seen whether an immune-reactivity to non-S1-RBD spike protein lowers the risk of contracting COVID-19 during the pandemic.

Our study has strengths and limitations. One strength is the population-based approach, covering 95% of the nation’s COVID-19 cases occurring during the first wave. In addition to one follow-up sample, we could study pan-SARS-CoV-2 S1-RBD antibody titers five months after symptom onset on the occasion of a second follow-up. Further, the assay’s specificity could be studied in a large population-based collective. A limitation of the study is that samples employed for the evaluation of specificity were selected from contemporary and not pre-pandemic participants. The samples, however, were taken during a time with a very low incidence of COVID-19, and positive samples were confirmed by taking a clinical history and a second antibody assay with an orthogonal testing algorithm. Then, we found some associations between pan-SARS-CoV-2 S1-RBD antibody titers and fever or smoking. Because we tested several associations, these findings may be prone to type I error. The fact that we could confirm the findings in the second follow-up sample and the fact that the associations are biologically plausible, however, makes such a type I error less probable. Finally, it would have been interesting to confirm the hypothesis that non-S1-RBD antibody reactive samples could confer immunological protection to a certain extent with neutralization assays. Unfortunately, we did not have the ability to test 248 samples with such an assay. In summary, however, we believe that the limitations do not invalidate our findings.

In conclusion, we demonstrated that the pan-SARS-CoV-2 S1-RBD Ig assay has favorable analytical, diagnostic and operational characteristics. An orthogonal testing approach helped to improve positive predictive values. We found that antibody-positive patients with SARS-CoV-2 infection have sustained levels of pan-SARS-CoV-2 S1-RBD antibody titers over nearly five months. We further found an association of the severity of disease with antibody levels and identified fever during COVID-19 as a determinant of pan-SARS-CoV-2 S1-RBD antibody titers. Interestingly, in patients with SARS-CoV-2 infection, smokers had lower antibody titers than non-smokers patients, despite a similar disease duration. Finally, in our population-based setting, the pan-SARS-CoV-2 S1-RBD assay helped to identify a substantial proportion of individuals without SARS-CoV-2 infection displaying antibody reactivity to non-S1-RBD spike protein epitopes, conferring potential clinical cross-reactivity e.g., with endemic common cold coronaviruses to SARS-CoV-2.

## Figures and Tables

**Figure 1 jcm-09-03989-f001:**
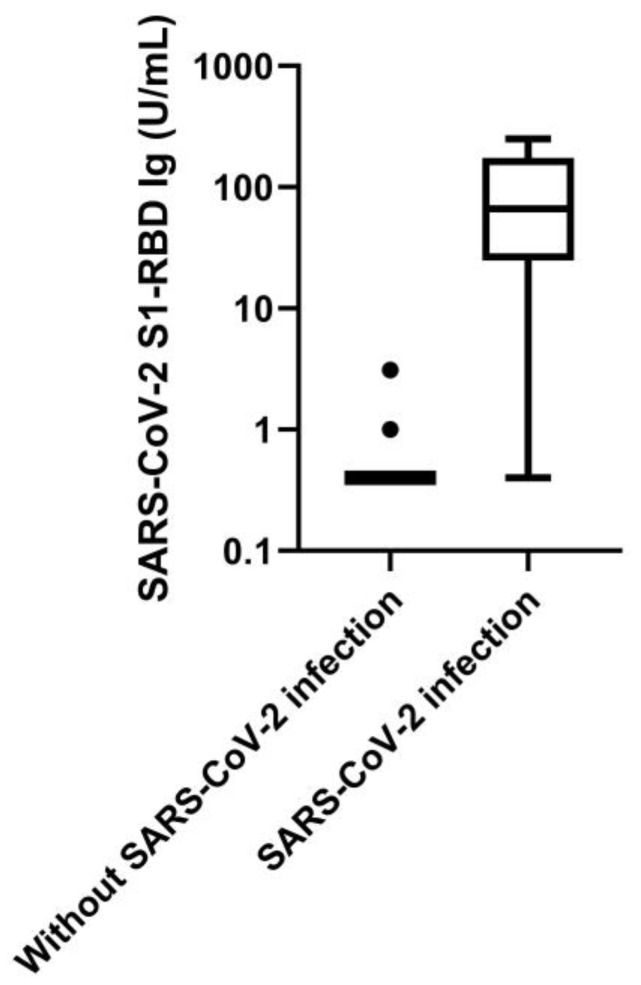
SARS-CoV-2 S1-receptor binding domain (RBD) Ig levels in patients with (*n* = 125) and without SARS-CoV-2 infection (*n* = 1159) on a logarithmic scale. Boxplots display medians and interquartile range; whiskers are shown according to Tukey’s.

**Figure 2 jcm-09-03989-f002:**
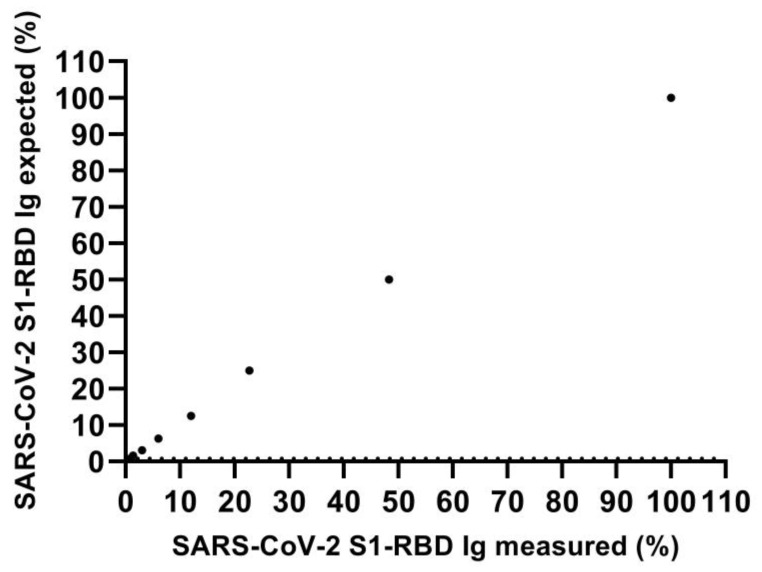
Pan-SARS-CoV-2 S1-RBD Ig levels upon serial dilution (1:2, 1:4, 1:8, 1:16, 1:32, 1:64 and 1:128) of a pool serum with a concentration of 105 U/mL are displayed. The dotted line displays the manufacturer’s cut-off (0.8 U/mL).

**Figure 3 jcm-09-03989-f003:**
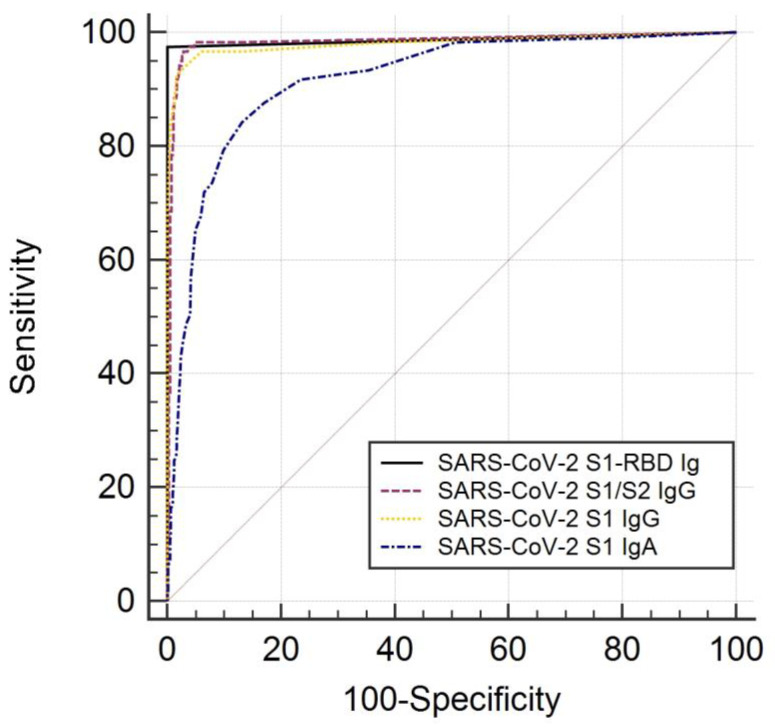
Receiver operating characteristic curves of different antibody assays measuring immunoglobulins directed against spike protein. The optimum decision point for SARS-CoV-2 S1-RBD antibodies was a cuttoff index COI > 1.0 U/L.

**Figure 4 jcm-09-03989-f004:**
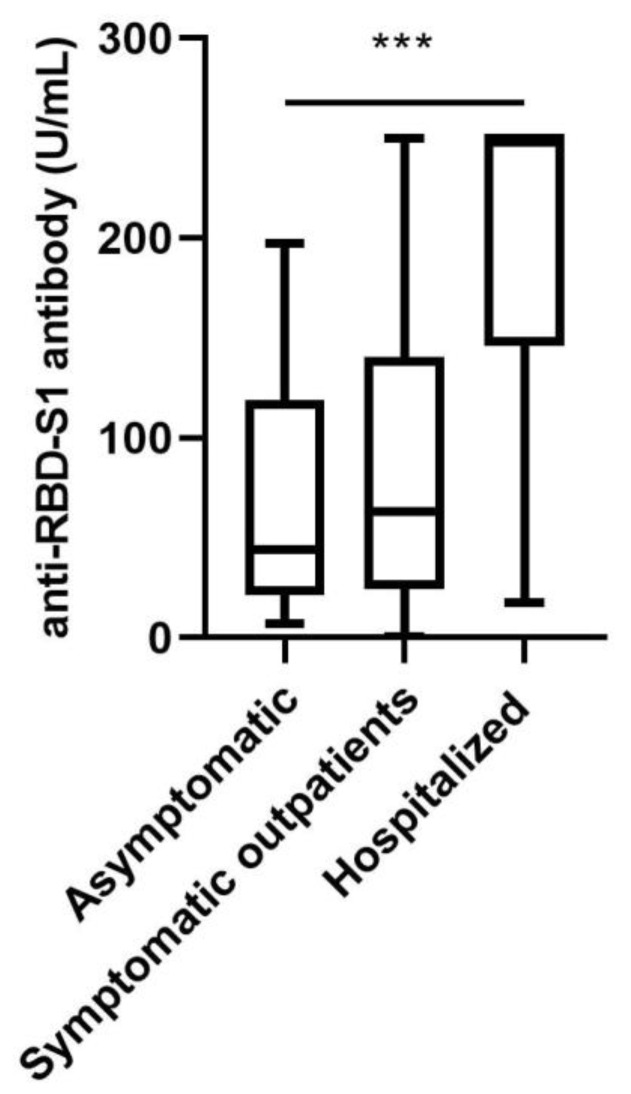
Pan-SARS-CoV-2 S1-RBD antibody titers in patients with SARS-CoV-2 infection stratified according to the severity of disease (hospitalized patients, *n* = 12; symptomatic outpatients, *n* = 104; asymptomatic individuals, *n* = 9). Medians and interquartile range are shown; whiskers are shown according to Tukey’s. *** *p* < 0.001.

**Figure 5 jcm-09-03989-f005:**
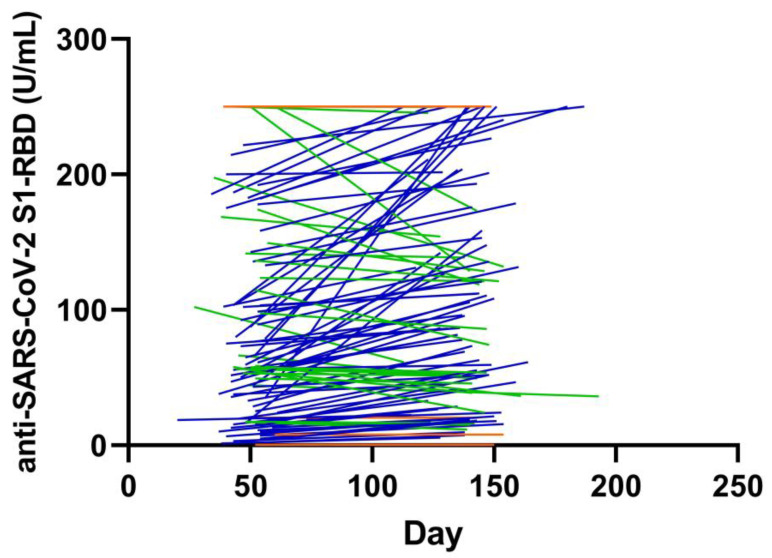
Course of antibody levels of pan-SARS-CoV-2 S1-RBD Ig in patients with SARS-CoV-2. Patients with ascending levels are shown in blue, patients with descending antibody levels are displayed in green, and patients with unchanged antibody levels are given in orange. In total, 15 patients had unchanged results at the upper limit of quantification at 250 U/mL.

**Figure 6 jcm-09-03989-f006:**
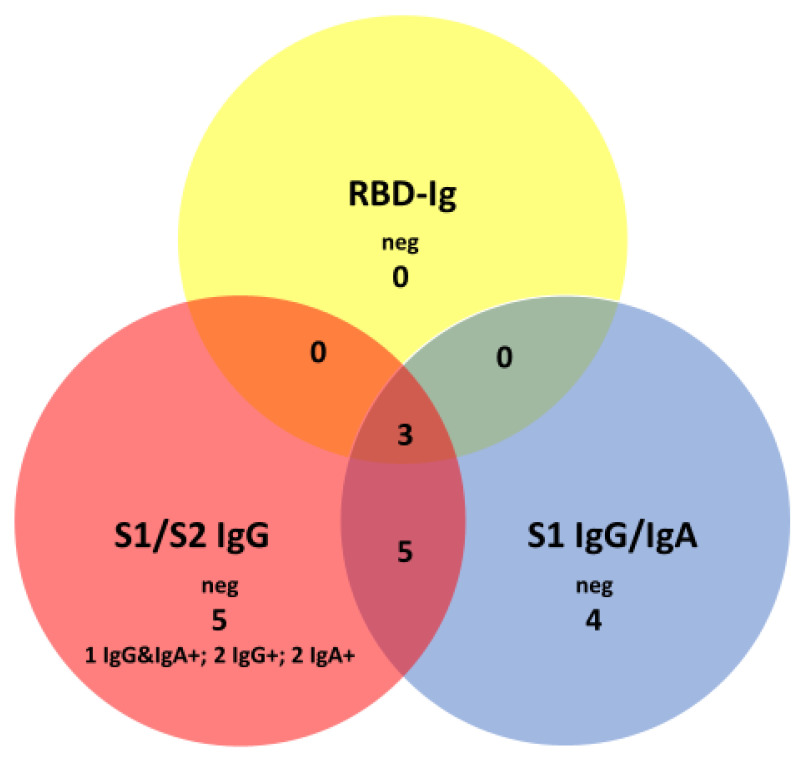
Venn diagram displaying negative anti-spike protein antibody assay results in 125 patients with SARS-CoV-2 infection.

**Figure 7 jcm-09-03989-f007:**
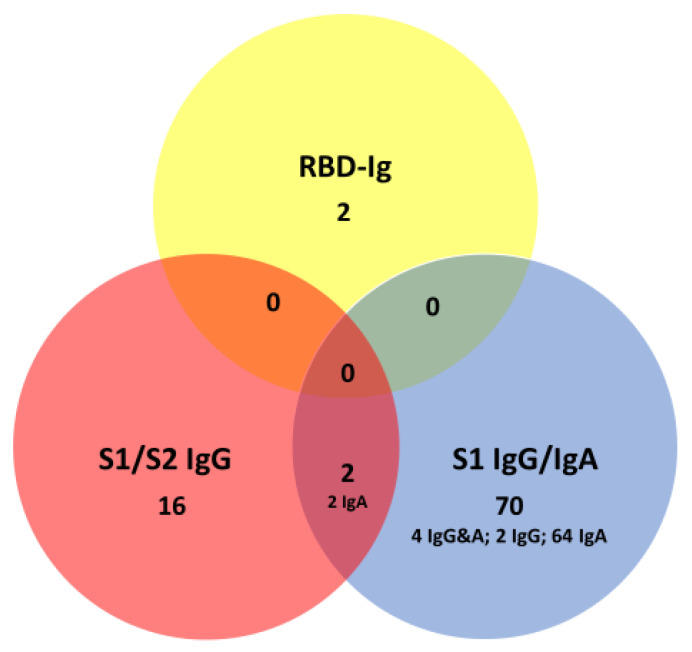
Venn diagram displaying positive anti-spike protein antibody assay results in 1159 individuals without SARS-CoV-2 infection. Manufacturers’ cutoffs were used as a criterion to adjudicate test positivity.

**Figure 8 jcm-09-03989-f008:**
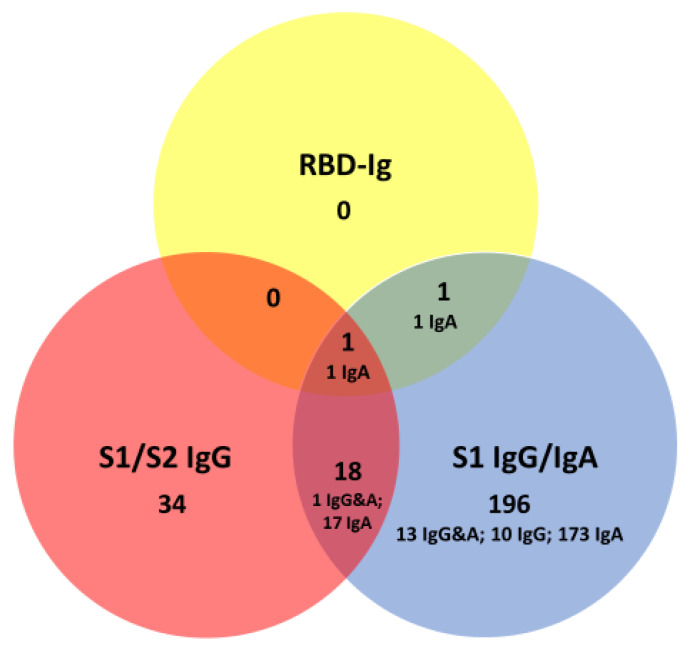
Venn diagram displaying positive anti-spike protein antibody assay results in 1159 individuals without SARS-CoV-2 infection. Half of the manufacturers’ cutoffs was used as a criterion to adjudicate test positivity.

**Figure 9 jcm-09-03989-f009:**
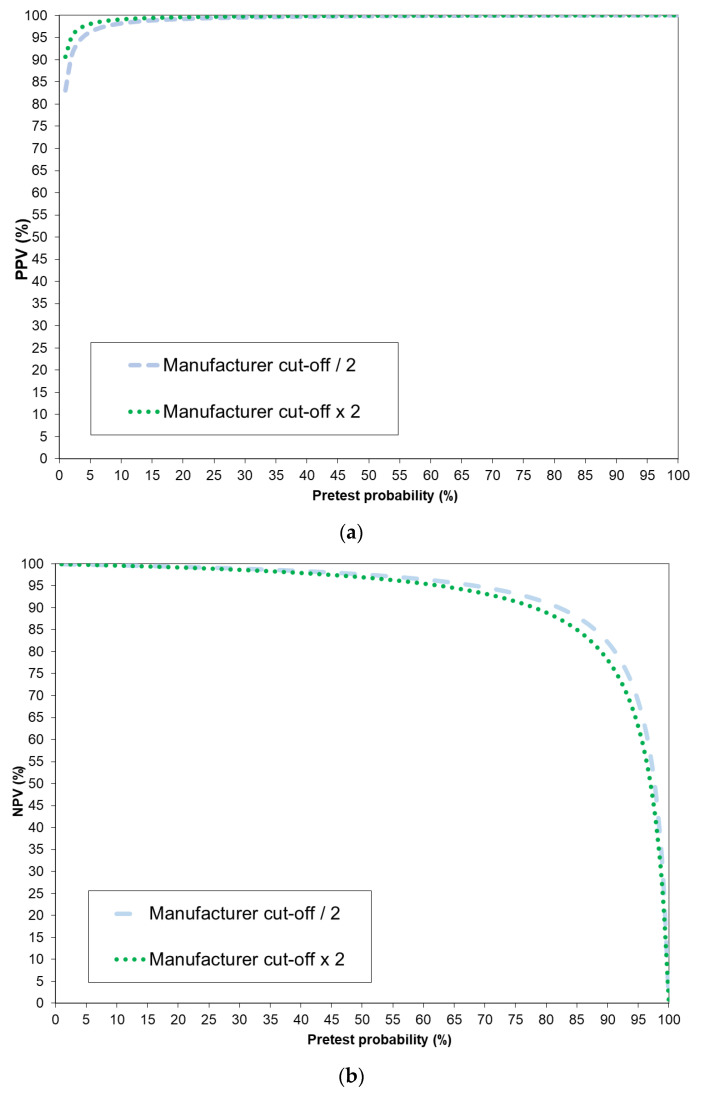
Positive (PPV; panel (**a**)) and negative (NPV; panel (**b**)) predictive values of pan-SARS-CoV-2 S1-RBD Ig at two different cut-offs (half and double of the manufacturers’ cutoff) over the whole range of possible pretest probabilities.

**Table 1 jcm-09-03989-t001:** Summary of sample types included into the study. N/A, not applicable; EBV, Epstein-Barr -Virus; CMV, cytomegalo virus; HCoV, endemic common cold human coronavirus; RT-PCR, real-time reverse transcriptase polymerase chain reaction.

Cohort	Patients/Participants	Criteria for Infection	Criteria for Absence of Infection	Evaluation of
COVID-FL	COVID-19 index cases & Infected close contacts (*n* = 125)	History and RT-PCR positive History and serology positive	N/A	Diagnostic sensitivity
COVID-FL	Non-infected close contacts (*n* = 123)	N/A	RT-PCR and/or serology negative	Diagnostic specificity
COVI-GAPP	Non-infected participants (*n* = 1036)	N/A	Absent history and serology negative	Diagnostic specificity
Biobank samples	Pre-pandemic patients with EBV (*n* = 8), CMV (*n* = 7), HCoV (*n* = 12)	Laboratory confirmed infection	N/A	Analytic specificity

**Table 2 jcm-09-03989-t002:** Pan-SARS-CoV-2 S1-RBD Ig antibody levels in patients with SARS-CoV-2 infection according to clinical symptoms during the course of the disease. Samples were taken at a median of 48 days after symptom onset. Mann-Whitney U test was used testing for statistical significance. Significant *p*-values are given in bold font.

Symptom	Antibody Level in Patients with Symptom, U/mL Median (IQR)	Antibody Level in Patients without Symptom, U/mL Median (IQR)	*p*-Value
Cough	79 (34–184)	54 (10–149)	0.17
(*n*)	(80)	(45)	
Fever	99 (53–205)	46 (17–120)	**0.001**
(*n*)	(61)	(64)	
Dysgeusia	62 (21–142)	71 (33–178)	0.58
(*n*)	(59)	(66)	
Headache	60 (20–189)	66 (35–138)	0.99
(*n*)	(60)	(65)	
Fatigue	73 (21–196)	63 (33–149)	0.75
(*n*)	(59)	(66)	
Anosmia	59 (25–118)	73 (24–193)	0.3
(*n*)	(48)	(77)	
Bone, joint and muscle pain	53 (17–119)	76 (27–182)	0.17
(*n*)	(41)	(84)	
Rhinitis	66 (22–128)	62 (25–182)	0.56
(*n*)	(40)	(85)	
Sore throat	79 (20–183)	79 (34–184)	0.81
(*n*)	(38)	(87)	
Chest pain	74 (25–250)	61 (23–166)	0.69
(*n*)	(34)	(91)	
Dyspnea	82 (22–250)	62 (27–144)	0.4
(*n*)	(28)	(97)	
Diarrhea	35 (13–128)	38 (74–175)	0.06
(*n*)	(27)	(98)	
Malaise	57 (14–116)	76 (27–180)	0.17
(*n*)	(25)	(100)	
Nausea	75 (41–99)	63 (24–180)	0.6
(*n*)	(13)	(112)	
Smoking	24 (7–80)	70 (30–178)	**0.05**
(*n*)	(11)	(114)	

**Table 3 jcm-09-03989-t003:** Predictive values (PV) for negative (NPV) and positive (PPV) results for the pan-SARS-CoV-2 S1-RBD Ig assay performed as a single test and in an orthogonal testing approach together with the pan-SARS-CoV-2 N Ig, both measured with electrochemiluminescence immunoassay (ECLIA). PV are shown in function of disease prevalence/pretest probability for SARS-CoV-2 infection.

Pretest Probability	S1-RBD-Ig PPV	S1-RBD-Ig NPV	S1-RBD-Ig and N-Ig PPV	S1-RBD-Ig and N-Ig NPV
40%	99.7%	98.4%	100%	7.1%
30%	99.5%	99%	100%	10.7%
25%	99.4%	99.2%	100%	13.3%
20%	99.2%	99.4%	100%	17%
15%	98.9%	99.6%	100%	22.5%
10%	98.2%	99.7%	100%	31.5%
8%	97.7%	99.8%	100%	37%
5%	96.3%	99.9%	100%	49.3%
3%	93.8%	99.9%	100%	62.3%
2%	90.9%	100%	100%	71.5%
1%	83.1%	100%	100%	83.5%

## Data Availability

The data used to support the findings in this study will be available from the corresponding author upon reasonable request.

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
