# Peer review of "Characterization of a Pan-Immunoglobulin Assay Quantifying Antibodies Directed against the Receptor Binding Domain of the SARS-CoV-2 S1-Subunit of the Spike Protein: A Population-Based Study"

_jcm, 2020, doi:10.3390/jcm9123989_

Round 1
Reviewer 1 Report
In their manuscript, “Characterization of a Pan-Immunoglobulin Assay Quantifying Neutralizing Antibodies Directed against the Receptor Binding Domain of the SARS-CoV-2 S1-Subunit of the Spike Protein: A Population-Based Study”, the authors evaluated the performance of the Roche Elecsys® Anti-SARS-CoV-2 S antibody assay. The authors use samples from SARS-CoV-2 infected individuals and those confirmed as antibody positive by two different assays as positive controls. Negative controls were obtained from a number of different sources, all individuals who were negative by RT-PCR for SARS-CoV-2 or antibody non-responsive by a two or more other assays. Overall this is a thorough and well written analysis of the Roche Elecsys® Anti-SARS-CoV-2 antibody assay. It is unfortunate that samples from early in infection were not tested to determine sensitivity by duration of infection, and pre-pandemic samples are missing for a more accurate estimation of specificity.
Major Issues:
- The title is misleading. This is an evaluation of the Roche Elecsys® Anti-SARS-CoV-2 antibody assay. No neutralizing antibody studies were performed. The word neutralizing needs to be removed from the title.
- SARS-CoV-2 infection and COVID-19 are used interchangeably which is incorrect. This needs to be corrected throughout the paper. In the abstract line 56 and 57, “…asymptomatic individuals with confirmed COVID-19”. The authors mean SARS-CoV-2 infection without symptoms. In line 171, the presence of COVID-19 is not based on RT-PCR, COVID-19 are the symptoms of SARS-CoV-2 infection. This line should state the presence of SARS-CoV-2 infection. Another example occurs on line 487, “…past COVID-19 infection.” Should be past SARS-CoV-2 infection. For this paper it is particularly critical as the authors to make this distinction between infection with the SARS-CoV-2 virus and the disease it causes, as they present antibody data from symptomatic (SARS-CoV-2 infected with COVID-19 disease) and asymptomatic (SAS-CoV-2 infected but no COVID-19 disease) individuals.
- The authors performed repeated testing on the same samples on multiple days (lines 197 to 206), but the amount of variation generated by the assay itself is not presented anywhere in the manuscript.
Minor issues:
- As there are multiple cohorts used for different purposes, a table describing all the samples used, their source, criteria for being considered SARS-CoV-2 infected or not, and purpose in the manuscript would be helpful.
- The number of subjects for each group presented in figure 4 should be stated. What is the sensitivity of symptomatically infected vs. asymptomatically infected individuals.
- The samples evaluated were tested previously by a host of different antibody assays (lines 148-161), why are the direct comparisons of these assays to the Roche Elecsys® Anti-SARS-CoV-2 antibody assay not presented in this manuscript?
- The authors performed ROC analysis (lines 305 to 317). Based on this analysis is the manufacturer’s cut-off of 0.8 U/m the best value to discriminate samples from SARS-CoV-2 infected vs. uninfected individuals?
Reviewer 2 Report
Overall: This manuscript describes good solid scientific investigation on a very topical area of research. My only criticisms is that it claims to show antibody neutralization, which it doesn’t. It shows antibody binding, with a functional biological test using native virus or pseudotype surrogates. In addition, the specificity of the RBD region in assays, to avoid cross reaction with our coronaviruses is known in the field, but the authors to do acknowledge this in the Intro or Discussion. This knowledge is presumably what lead the company (unnamed) supplying the ECLIA assay to design it this way.
Title misleading. The ECLIA and other tests detailed measure binding antibodies NOT neutralization. Remove neutralization from title and L486. In contrast ref 43 (L494 does show neut using PVs) and indeed on L510 the authors state they didn’t have time to performs neuts
Abstract (L45)
Authors should briefly mention positivity test used on the 125 in Abstract and Methods, not just refer to other publications. (historical samples tested are by IgG and IgM assay – ELISA?). Cohort 2 patients were ‘asked’ if they had tested RT-PCR CoV2 positive – was this really just question to participants or was proof obtained?
- Intro (Line 70)
Very clear, describing RT-PCR testing and role of antibody assays in clarifying diagnosis. Lack of info on RBD and specificity finding in other studies.
Line 91 – clarify where combined immunoglobulin test and individual.
Line 102 – who manufactures the ECLIA? – Roche, the same as the anti-N on Line 153?
- Methods (L108)
Line 133 – clarify endemic coronavirus disease – presume seasonal cold CoV as at least a week before first confirmed COVID19 case in area – see later para L287. How was positivity of sera determined??
Line 171 – does this refer to both patient cohorts? So PCR negative but antibody positive was considered COVID19 positive?
- Lab analysis (L183)
- Results (L231)
L280 – how do they know which particular seasonal CoV the patients had from serum?
L281 – no seasonal CoV patients showed positive in ECLIA
L298 – typo ‘had was’
Section 4.6 – maybe also give % incr, decr and same?
Venn diagrams are a useful and clear depiction of data
Table 2 legend needs clarification – explain Preset Probs in terms of disease prevalance
- Discussion (L418)
Little information on other studies which demonstrated SARS-CoV2 specificity (cf seasonal corona for example) of RBD compared with other regions of the Spike
L432 – clarify sentence on measurement
L466 – clarify ‘excluding a past COVID19 infection’
L470 – associated – maybe correlated?
L492 – presumably manufacturers set cut offs to avoid this
L518 – lower not fewer
L521 – clarify cross reactivity – meaning with seasonal coronaviruses?
Author Response
Pleas see the attachment
